# A DFT Approach to the Surface-Enhanced Raman Scattering of 4-Cyanopyridine Adsorbed on Silver Nanoparticles

**DOI:** 10.3390/nano9091211

**Published:** 2019-08-28

**Authors:** Isabel López-Tocón, Samuel Valdivia, Juan Soto, Juan Carlos Otero, Francesco Muniz-Miranda, Maria Cristina Menziani, Maurizio Muniz-Miranda

**Affiliations:** 1Andalucía Tech, Unidad Asociada IEM-CSIC, Departamento de química Física, Facultad de Ciencias, Universidad de Málaga, 29071 Málaga, Spain; 2Present Address: Chimie ParisTech, PSL Research University, CNRS, Institute of Chemistry for Life and Health Sciences, F-75005 Paris, France; 3Department of Chemical and Geological Sciences, University of Modena and Reggio Emilia, Via Campi 103, 41125 Modena, Italy; 4Department of Chemistry “Ugo Schiff”, University of Florence, Via della Lastruccia 3, 50019 Sesto Fiorentino, Italy

**Keywords:** SERS, silver sol, metal nanoparticles, DFT calculations, 4-cyanopyridine

## Abstract

A Surface-Enhanced Raman Scattering (SERS) spectrum of 4-cyanopyridine (4CNPy) was recorded on silver plasmonic nanoparticles and analyzed by using Density Functional Theory (DFT) calculations. Two simple molecular models of the metal–4CNPy surface complex with a single silver cation or with a neutral dimer (Ag^+^–4CNPy, Ag_2_–4CNPy), linked through the two possible interacting sites of 4CNPy (aromatic nitrogen, N, and nitrile group, CN), were considered. The calculated vibrational wavenumbers and intensities of the adsorbate and the isolated species are compared with the experimental Raman and SERS results. The analysis of the DFT predictions and the experimental data indicates that 4CNPy adsorbs preferentially on neutral/charged active sites of the silver nanoparticles through the nitrogen atom of the aromatic ring with a perpendicular orientation.

## 1. Introduction

Surface-Enhanced Raman Scattering (SERS) is becoming a versatile spectroscopic technique in different scientific fields, such as analytical chemistry, for the detection of analytes/contaminants [1,2] at trace levels, even reaching single-molecule detection [3], biochemistry and forensic medicine, [4,5,6] for the detection of RNA by the recognition of adenine and cytosine using silver nanoparticles, pharmaceutical medicine [7], in the study of new target drugs, and other applications in restoration and artistic heritage [8], archeology [8], food [9,10], etc.; that is, most SERS studies deal with the molecular recognition of vibrational fingerprints of molecules at low concentration. This is due to the main characteristics of the SERS phenomenon: The enormous enhancement of the molecular Raman signal in the vicinity of a rough metal surface of nanometric size [11,12], its high selectivity, and sensitivity.

Since SERS is a technique where the metal–solution interface plays an important role, it is employed in the study of electrochemical processes [13], such as heterogeneous catalysis, corrosion processes, and electrochemical reduction of molecules like 4-cyanopyridine (4CNPy). Electrochemical SERS experiments of 4CNPy on silver electrodes have been extensively studied [14,15,16,17] under different experimental conditions varying the concentration, the electrode potential, or the type of solvent or electrolyte, because different pyridine-like species can be adsorbed on the surface due to the reduction process of the nitrile group. This yields mixed-up SERS spectra in which appear new bands different to those of the Raman of solid 4CNPy or the aqueous solution, for example at 625, 1580, and 2103 cm^−1^ [16]. Therefore, the analysis of electrochemical SERS spectra of 4CNPy becomes a difficult task. Colloidal SERS spectra of 4CNPy have been less studied [18,19]. Even these spectra are simpler than those recorded on electrode, and the reduction of the nitrile group shows metal selectivity being observed only on copper surfaces [19].

In addition, the analysis of SERS spectra of 4CNPy in both electrode and colloidal experiments has been carried out by identifying new bands appearing in the spectra or by evaluating the vibrational wavenumber shifts or the change in the relative intensity of the nitrile band. Few theoretical studies modelling the metal–4CNPy surface complex have been published. Only ab initio calculations of 4CNPy adsorbed on lattice points of silver crystals have been reported by Osaki el al. [18] in order to estimate the vibrational wavenumbers at the minimum HF/3-21G level.

We report here the whole SERS spectrum of 4CNPy recorded on silver nanoplasmonic particles at low concentration in order to guarantee the adsorption of the monomeric species, thereby avoiding aggregation. Furthermore, the addition of the electrolyte on silver colloid is studied. The SERS spectra have been analyzed on the basis of DFT calculations of simple linear models of the metal–adsorbate surface complex, where 4CNPy is bonded to a single silver cation (Ag^+^–4CNPy) or to a neutral dimer (Ag_2_–4CNPy) through the two possible interacting sites, the aromatic nitrogen and nitrile group. This computational approach is consistent with the chemical enhancement mechanism of the SERS effect based on the adatom model [20,21,22]. In this model, the interaction of the adsorbed molecules occurs with surface defects constituted by one or a few metal atoms, which can be considered almost isolated on the metal. The validity of the adatom approximation was widely verified for many adsorbed molecules by DFT calculations that allowed us to satisfactorily reproduce the corresponding SERS spectra [23,24,25,26,27,28,29], including band positions and relative intensities, as well as to understand the type and strength of the molecule/metal interactions. The calculated vibrational wavenumber shifts with respect to the isolated molecule and the calculated Raman intensities of the different surface complexes are compared with the experimental results. This allows for deducing the adsorption center and the molecular orientation of 4CNPy on the silver surface as well as the effect of the adsorption on the electronic structure of the adsorbate related to their vibrational properties and geometry, which are useful data for modifying the electroluminiscent performance of 4CNPy in Organic Light-Emitting Diodes (OLEDs), for instance, since 4CNPy is used as n-type unit to build bipolar host materials [30].

## 2. Materials and Methods 

### 2.1. Experimental Section

Silver hydrosols were prepared by following the Creighton’s procedure [31], by adding AgNO_3_ (99.9999% purity, Aldrich, Spain) to excess NaBH_4_ (99.9% purity, Aldrich, Spain) and waiting a week before using them for SERS measurements in order to avoid the presence of residual reductant [32]. This delay provokes a broadening of the plasmonic band at 394 nm towards longer wavelengths (see Appendix A, black line). NaCl was added to a portion of the Ag colloid in 10^−3^ M concentration, in order to improve SERS enhancement. This addition promotes a redispension of the silver nanoparticles (see Appendix A, red line), which can be considered mono-dispersed, as shown in the TEM image of Appendix A, with sizes ranging between 5 and 20 nm. However, the occurrence of a small band around 570 nm could be attributed to a minority of silver aggregates. Then, 4CNPy was added to both the colloidal samples, with 10^−4^ M final concentration. At this concentration, no effects of dimer or aggregated surface species were detected [14], although different researchers differ on the molecular orientation of 4CNPy on silver surface [14,17].

Raman spectra of solid 4CNPy and 0.7 M aqueous solution of 4CNPy were obtained, as well as SER spectra of 4CNPy in Ag colloids, with and without chloride anions.

All Raman and SERS spectra were recorded using the 514.5 nm line of a Coherent argon ion laser, and a Jobin–Yvon HG2S monochromator equipped with a cooled RCA-C31034A photomultiplier (Horiba-Jobin-Yvon, Japan). A defocused laser beam with 100 mW power was employed for impairing thermal effects. Power density measurements were performed with a powermeter instrument (model 362; Scientech, Boulder, CO, USA) giving about 5% accuracy in the 300–1000 nm spectral range. 

The UV-visible absorption spectra of the colloid suspensions diluted to 10% in ultra-pure water (Milli-Q system, 18.2 MΩ cm resistivity) to avoid spectral saturation were recorded in a Cary 5 double-beam spectrophotometer (Agilent, CA, USA), using a 1 cm pathlength cell.

Transmission electron microscopy (TEM) on Ag colloids was obtained by using a a Philips EM 201 instrument (ThermoFisher Scientific, MA, USA) with an electron beam emitted at 80 kV, after placing a drop of colloidal sample on a carbon–Cu grid.

### 2.2. Computational Details

Simple molecular models for the Ag–4CNPy surface complex were assumed involving one and two silver atoms (Ag^+^–4CNPy and Ag_2_–4CNPy), respectively, linked through the two possible interacting sites of 4CNPy (aromatic nitrogen, N, and cyano group, CN). A silver cation (Ag^+^) and the neutral dimer (Ag_2_^0^) were selected to simulate the density of charge of the colloidal surface. 

B3LYP [33,34] functional with the LanL2DZ [35] basis set was employed for calculating the respective optimized structures and force fields of isolated 4CNPy and the different complexes (Ag^+^–N(4CNPy); Ag^+^–CN(4CNPy); Ag_2_–N(4CNPy); and Ag_2_–CN(4CNPy). This level of calculation has been previously used by us for analyzing the SERS spectra of pyridine [36] or similar benzene-like molecules [37,38]. Geometry optimizations were constrained to a planar structure under C_2v_ symmetry and all the calculated wavenumbers of the discussed complexes are real, indicating that the optimized geometries correspond to equilibrium structures. All the calculations were carried out using the GAUSSIAN 09 package [39].

The Raman intensities were calculated with the following Equation [40]: (1)$$I_{k}=\frac{d\sigma }{d\Omega }=\frac{\pi^{2}}{\varepsilon_{0}^{2}}{\left({\tilde{\nu }}_{1}-{\tilde{\nu }}_{k}\right)}^{4}\frac{h}{8\pi^{2}c{\tilde{\nu }}_{k}}\left(S_{k}/45\right)\frac{1}{1-\exp\left(hc{\tilde{\nu }}_{k}/k_{B}T\right)},$$

Where the intensity of k-th mode is given by the differential cross section (cm^2^/sr), *S_k_* is the scattering factor (Å^4^/amu) calculated with the polarizability gradient method by GAUSSIAN 09:(2)$$S_{k}=45{\left(\frac{d\alpha }{dQ_{k}}\right)}^{2}+7{\left(\frac{d\gamma }{dQ_{k}}\right)}^{2},$$
*ε_0_* is the permittivity of vacuum, *c* is the speed of light, *h* is the Planck constant, *k_b_* is the Boltzmann constant, *T* is the temperature, ${\tilde{\nu }}_{1}$ is the wavenumber of the incident light, ${\tilde{\nu }}_{k}$ is the wavenumber of the respective k-th vibrational mode.

## 3. Results and Discussion

### 3.1. Raman and SERS Spectra of 4CNPy. Vibrational Assignment

Raman spectra of solid 4CNPy and its aqueous solution (Figure 1) are dominated by four strong bands recorded at about 2250, 1600, 1200, and 1000 cm^−1^, which are assigned to the totally symmetric modes ν(CN), 8a;ν_ring_, ν(CX), and 12;δ_ring_, respectively (Appendix A). However, the relative intensity of these four bands is slightly different in these two spectra, recorded at 991 cm^−1^, the strongest band in the Raman of the solid, while the 2250 cm^−1^ line dominates the spectrum of the aqueous solution.

These four bands are also strong in the SERS spectra, recorded with or without chloride ions in the colloidal bulk. It should be noted that, in both cases, the intensity of the band recorded at 2250 cm^−1^ becomes weaker than that recorded in the normal Raman spectra. Furthermore, the relative intensities of the remaining SERS bands with chloride show similar intensity to the mid-wavenumber region of Raman of the solution. The band recorded at about 1000 cm^−1^ is the strongest one, while the SERS recorded without chloride is now dominated by the band recorded at 1600 cm^−1^. The presence of the CN stretching band (2250 cm^−1^) in the two SERS spectra and other weak CN-sensitive bands, recorded at about 566 and 784 cm^−1^ and denoted as δ(CCN) and 1,ν_ring_+ν(CN), respectively, together with the fact that no decomposition bands appear, implies that no chemical reduction of the nitrile group due to the excess of borohydride in the bulk takes place. Moreover, SERS of 4CNPy shown in Figure 1 corresponds to a monomer species, as expected for this low-concentration spectra [14], given that the dimer or aggregated species are characterized by Raman bands at 1520, 1261, and 970 cm^−1^, which are missing in our records.

The strong decreasing of the intensity of the CN stretching band in SERS spectra has been attributed in previous work [15,16,41,42] to three different effects: To the relative orientation of the CN group with respect to the silver surface, depending on the position of the nitrile group in different cyanopyridines, being especially striking in the case of 2CNPy [15], to a tilted orientation of the adsorbate in the cases of 3CNPy and 4CNPy [16], or to the disappearance of the CN group due to the formation of the corresponding amide [41,42]. According to the surface propensity rules derived from the electromagnetic mechanism of SERS [11,43,44], a tilted orientation would imply an enhancement of out-of-plane normal modes. In our spectra, the chemical transformation of the nitrile group to the corresponding amide is discarded. No strong bands assigned to out-of-plane are enhanced, and all the SERS bands recorded with strong or medium intensities are assigned to in-plane fundamentals, almost all of them corresponding to totally symmetric modes (Appendix A). Therefore, 4CNPy should be adsorbed on the silver surface with perpendicular orientation. Appendix A collects the Raman and SERS experimental wavenumbers and the proposed vibrational assignment of 4CNPY on the basis of the previously reported force field for the isolated 4CNPy [14,45] and the correlation with the results of pyridine [46]. The calculated B3LYP/LanL2DZ wavenumbers agree quite well with the experimental values of the solid and the aqueous solution. This level of calculation reproduces the red-shifts of the wavenumbers of modes 1 and 6a, recorded at 784 and 468 cm^−1^, respectively, in the solution Raman spectrum, with respect to those in pyridine (1005 and 618 cm^−1^, respectively), as expected in monosubstituted benzene-like derivatives [47,48]. On the other hand, the presence of chloride in the colloidal solution induces an overall enhancement of the SERS spectrum with well-defined bands, as can be seen when it is compared with the recorded without chloride, as a result of the well-known process of chloride activation [49,50,51]. Actually, chloride anions added to our silver colloids are strongly adsorbed on the metal surface, provoking a marked particle dispersion due to the repulsion between the negative charges (Appendix A), along with the formation of active sites (adatoms) on the silver surface able to give rise to metal/ligand complexes with strong SERS enhancement.

### 3.2. DFT Calculations. SERS Wavenumber Shifts

The optimized geometries and the respective calculated vibrational wavenumbers for isolated 4CNPy and for the selected models of the surface complex are collected in Appendix A. No scale factor is applied to the calculated wavenumbers. The molecular structure of 4CNPy in the two Ag–CN complexes does not change significantly with respect to the isolated species, and only a little shortening of the CN bond (from 1.1821 to 1.1758 Å) is predicted, while the remaining structural parameters keep almost insensitive. However, the two Ag–N complexes show some changes in the distances and angles of the aromatic ring resembling the shape of the 8a ring stretching normal mode, while the CN bond keeps its distance. This means than the geometry of 4CNPy is deformed towards a quinonoid structure when this molecule is adsorbed through the nitrogen, by shortening the intermediate CC bonds (from 1.4061 to 1.3989 Å) and by stretching the CC and CN distances (from 1.4136 to 1.4140 Å, and from 1.3578 to 1.3671 Å, respectively). The calculated energy difference between their respective Ag^+^–4CNPy optimized structures (Ag^+^–CN and Ag^+^–N) is 2.40 Kcal/mol. This value is reduced to 2.03 Kcal/mol in the case of neutral Ag_2_–4CNPy complexes (Ag_2_–CN and Ag_2_–N), being the most stable one in both cases when the coordination with the metal takes place through the aromatic nitrogen.

The experimental and B3LYP/LanL2DZ wavenumber shifts of the strongest Raman and SERS bands are shown in Table 1. Although the intense CN band recorded at 2250 cm^−1^ shows a small red-shift, the adsorption produces a general blue-shift of the vibrational wavenumbers. For instance, experimental shifts of 6, 4, and 6 cm^−1^ are observed for the most intense bands assigned to 8a, ν(CX), and 12 fundamentals, respectively. These small vibrational shifts are similar to those recorded in the electrochemical SERS spectra of pyridine [36].

This general behaviour is roughly reproduced by the calculated results predicted for the two Ag–N complexes, which show, for the abovementioned normal modes, overestimated wavenumber shifts of 15, 3, and 23 cm^−1^, respectively, in the case of Ag_2_–N or even larger, 28, 10, and 40 cm^−1^, respectively, when the molecule is bonded to a single silver cation with a larger positive density of charge (Ag^+^–N). The overestimation of the calculated wavenumber shifts upon adsorption on neutral silver clusters has been already reported and can be corrected by a factor of 0.66 at this level of theory [36], which improves the agreement between experimental and calculated results. However, the calculated wavenumber shifts of the two Ag–CN complexes just predict the opposite trend to the experiment values. Red-shifts of −20, −2, and −4 cm^−1^ are calculated for this set of fundamentals in the case of the Ag^+^–CN complex. Therefore, the analysis of the SERS wavemumber shifts discards this kind of surface complexes, where 4CNPy is bonded to silver through the nitrile functional group.

This scaling way, the vibrational shifts for the Ag_2_–N complex not only fit better with the experimental ones, but also predict the experimental behaviour toward a blue-shifted wavenumbers, and it is the one that estimates small vibrational shift for the CN stretching, as shown experimentally.

### 3.3. DFT Calculations. Raman Intensities of Isolated 4CNPy or Bonded to Silver Clusters

Mid-wavenumber region of calculated B3LYP/LanL2DZ Raman spectra of isolated 4CNPy and of the selected silver complexes are shown in Figure 2 and Appendix A, together with the experimental records. The whole spectral range is drawn in Appendix A. The performance of the theoretical method in order to estimate the normal Raman intensities was checked by comparing the calculated relative intensities for isolated 4CNPy and those of the Raman of the aqueous solution. Table 2 collects the experimental and calculated intensity ratio for the strongest bands of isolated 4CNPy and the surface complexes with respect to the 1004 cm^−1^ line assigned to the 12; δ_ring_ normal mode. This band was selected as a reference because it is not selectively enhanced by metal–molecule charge transfer processes, as demonstrated in the analysis of the SERS spectra of pyridine [52]. The relative intensities of ν(CN), 8a, and ν(CX) strong bands are 1.37, 0.51, and 0.84, respectively, measured in the aqueous solution Raman spectrum, and amount to 2.66, 0.77, and 0.68 in the calculated results, respectively. The relative intensities of the mid-region of the Raman spectrum are well reproduced theoretically, although the relative intensities calculated for 8a and ν(CX) modes appear reversed. It must be noted that the calculated intensity for the ν(CN) band is twice the experimental intensity, which also happened for the 6a mode.

The calculated spectra for the surface complexes show significant changes in the relative intensity of the bands, depending on the surface charge and the type of metal–molecule bonding. For the Ag_2_–N neutral complex, the strongest band in the mid-wavenumber region is predicted at 1600 cm^−1^ (8a mode) with an intensity ratio of 1.70, while in the case of the Ag_2_–CN complex, the band at about 470 cm^−1^ (6a mode) also shows a very strong enhancement with the same ratio of 1.70. The positive-charged Ag^+^–4CNPy complexes show a different behavior to the neutral ones. The strongest band corresponds to the 1200 cm^−1^ line (ν(CX) mode) when the molecule is coordinated through the aromatic nitrogen, Ag^+^–N, with an intensity ratio of 4.80 duplicating that of the 8a mode (2.29). When 4CNPy is coordinated through the nitrile group, Ag^+^–CN, the 8a, ν(CX) and 12 bands have similar relative intensities (1.18, 0.95, and 1.0, respectively), while the 6a band calculated at 470 cm^−1^ also shows a strong intensity. 

Regarding the region beyond 2000 cm^−1^, the intensity of the CN stretching band always dominates any kind of calculated spectra (Table 2, Appendix A), as it was in the calculated Raman spectra of the isolated molecule. Intensity ratios of 11.54 and 11.30 are calculated for both Ag^+^–N and Ag_2_–CN complexes, respectively, which reduce to 5.73 and 2.33 in the Ag^+^–CN and Ag_2_–N models, respectively. Taking into account that the intensity of this band is overestimated by a factor of 2.0 in the calculated spectrum of the isolated molecule, the ratio of the ν(CN) band in the Ag_2_–N complex is being reduced to 1.16 and the band corresponding to the 8a fundamental should become the strongest one. If this scaling is applied to the calculated intensities, it can be concluded that only the results obtained for the Ag_2_–N complex are able to reproduce the experimental intensities of the SERS spectra obtained without chloride ions.

The effect of the molecular adsorption produces not only changes in the geometry, as has been previously discussed, but also in the charge distribution of the different complexes. Appendix A collects the B3LYP/LanL2DZ Mulliken’s charges. An amount of charge of −0.15 and −0.25 from the molecule to silver atoms is transferred in any neutral (Ag–4CNPy) and positive (Ag^+^–4CNPy) surface complexes, respectively, which is just related to the adsorption strength and the Ag–N or Ag–CN distances of about 2.40 and 2.15 Å, respectively (see Appendix A). All this reflects the effect of the metal on the electronic structure of the molecule [53].

To summarize, the analysis of the SERS vibrational shifts and intensities points to a surface complex where 4CNPy is bonded to neutral silver atoms of the nanoparticle through the aromatic nitrogen. Although the relative intensities of the SERS spectra recorded with or without chloride ions are similar, some changes can be highlighted. The main differences are related to the overall intensity of the spectrum and the relative intensities of modes 8a and 6a. The SERS with chloride ions is stronger with a larger signal/noise ratio, and their relative intensities resemble those of the Raman of the solution. The SERS obtained without chloride is characterized by the relative enhancement of the 1600 cm^−1^ band, corresponding to mode 8a and the very weak intensity of mode 6a. We have previously demonstrated that the selective enhancement of the 8a fundamental is related to the presence of resonant metal-to-molecule charge transfer (CT) processes in the SERS of aromatic molecules like pyridine and derivatives [52,53,54,55,56,57]. This CT mechanism of SERS is very dependent on the particular experimental conditions, mainly on the energy of the exciting line and the density of charge of the metal and the electric potential of the double layer. At a fixed laser line, the resonant CT condition is controlled by the electric properties of the metal surface and the interface [53,58,59], which are both dependent on the presence or lack of chloride, which is strongly adsorbed on silver. The differences between the selective enhancement of the bands of both SERS indicate that the CT contribution to the SERS of this molecule is larger in the SERS recorded without chloride, given that mode 8a dominates the spectrum. This means that chloride anions are preferably adsorbed on surface locations where the CT process is more favorable.

Finally, it is important to point out that chloride anions added to Ag colloidal suspensions are strongly bound to silver, but they do not impair the adsorption of organic molecules like 4CNPy, because they instead allow more effective molecular/metal interactions, promoting the formation of surface active sites (adatoms) on the metal surface, which are able to bind the ligand molecules.

## 4. Conclusions

SERS spectra of 4CNPy recorded on silver plasmonic nanoparticles at 10^−4^ M concentration correspond to the monomer. No dimer or aggregate species nor reduction of nitrile group due to an excess of borohydride is detected. The shifts of the vibrational wavenumbers and the relative intensities observed in the SERS spectra were analyzed. SERS spectra recorded without chloride ions are characterized by relative enhancement of the band recorded at about 1600 cm^−1^, while this band becomes weaker in the SERS recorded with chloride ions. DFT calculations on different charged silver surface complexes point out that the 4CNPy adsorbs through the aromatic nitrogen atom to neutral silver atoms in a perpendicular orientation, given that the Ag_2_–N surface complex is able to account for the vibrational wavenumber shifts and the relative intensities of the SERS bands. 

## Figures and Tables

**Figure 1 nanomaterials-09-01211-f001:**
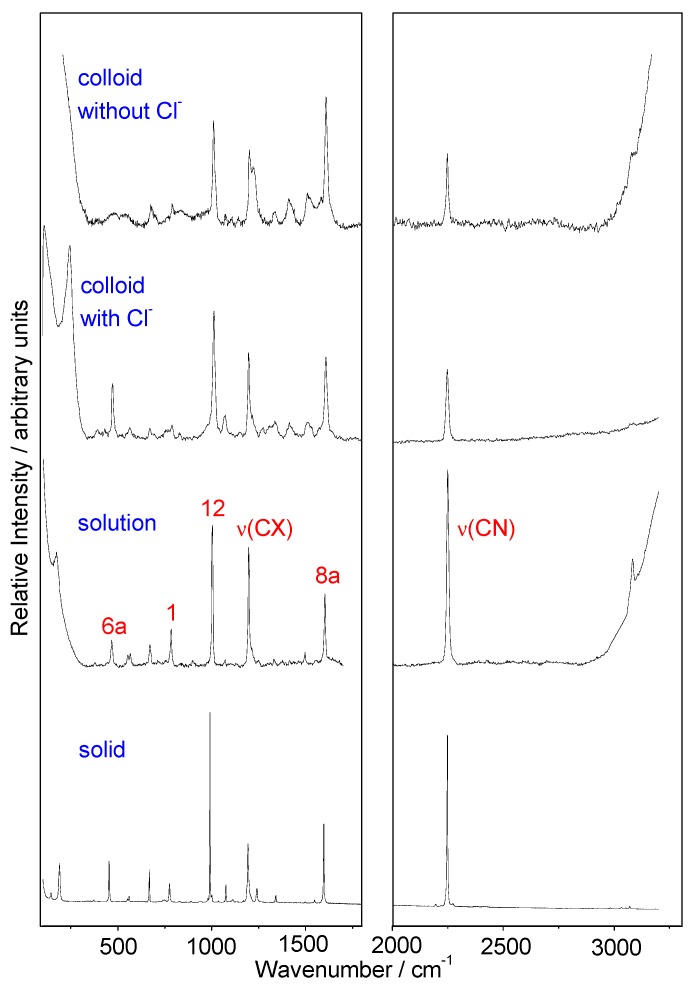
Raman spectra of 4-cyanopyridine (4CNPy) as solid sample and dissolved in water (0.7 M concentration), along with the Surface-Enhanced Raman Scattering (SERS) spectra of 4CNPy (10^−4^ M concentration) in Ag colloidal suspensions with or without addition of chloride ions. Excitation: 514.5 nm.

**Figure 2 nanomaterials-09-01211-f002:**
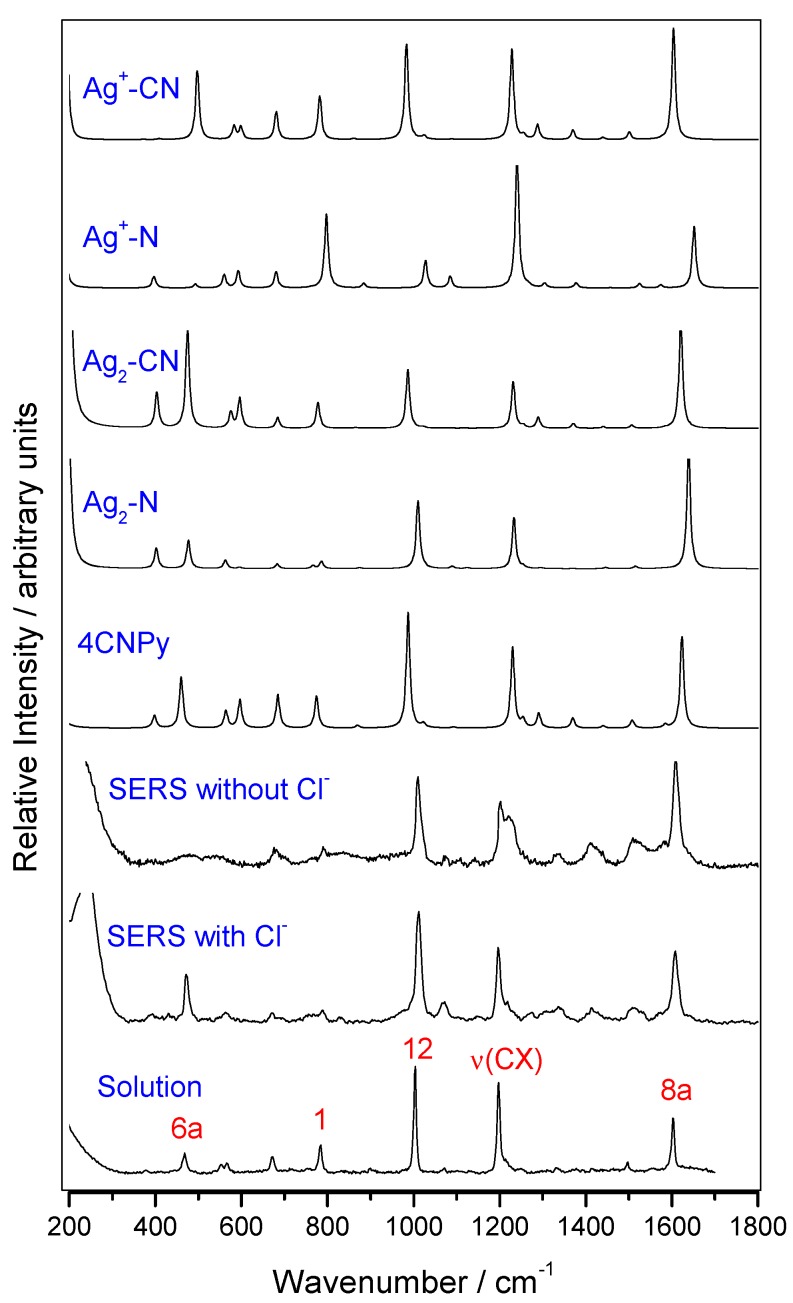
Mid-wavenumber region of the Raman and SERS spectra of 4CNPy and B3LYP/LanL2DZ calculated spectra for isolated 4CNPy and different surface complexes.

**Table 1 nanomaterials-09-01211-t001:** Experimental and calculated B3LYP/LanL2DZ vibrational wavenumbers (cm^−1^), and wavenumber shifts with respect to the Raman spectrum or to the calculated ones for isolated 4CNPy, respectively.

.	Experimental			Calculated B3LYP/LanL2DZ				
Assignment	Solution	SERS ^1^	SERS ^2^	4CNPy ^3^	Ag_2_–N	Ag^+^–N	Ag_2_–CN	Ag^+^–CN
2, ν(CH);A_1_	3084	4	-	3248	7	18	1	5
ν(CN);A_1_	2250	−2	−2	2271	6	13	26	26
8a, ν_ring_;A_1_	1602	6	6	1623	15	28	−3	−20
ν(CX), ν_ring_+ν(CN);A_1_	1198	4	−2	1230	3	10	1	−2
18a, δ(CH);A_1_	1072	-	0	1092	−3	−7	0	−3
12;δ_ring_;A_1_	1004	6	8	987	23	40	−1	−4
1, ν_ring_+ν(CN);A_1_	784	6	6	774	12	23	4	8
6a, δ_ring_;A_1_	468	-	4	460	16	33	15	37
3, δ(CH);B_2_	1330	4	6	1369	1	8	1	1
6b, δ_ring_;B_2_	672	4	0	684	−1	−4	0	−4
δ(CCN);B_2_	566	-	−4	563	−1	−3	12	20

^1,2^: Vibrational shifts measured in the SERS spectra recorded without and with chloride ions, respectively. ^3^: Calculated wavenumbers of isolated 4CNPy.

**Table 2 nanomaterials-09-01211-t002:** Experimental and calculated intensity ratio of the strongest bands of isolated 4CNPy and its surface complexes with respect to the band assigned to mode 12.

		Experimental			Calculated B3LYP/LanL2DZ				
Modes	ν (cm^−1^)	Sol. ^1^	SERS ^2^	SERS ^3^	4CNPy	Ag_2_-N	Ag^+^-N	Ag_2_-CN	Ag^+^-CN
ν(CN);A_1_	2250	1.37	0.70	0.57	2.66	2.33	11.54	11.30	5.73
8a, ν_ring_;A_1_	1602	0.51	1.22	0.63	0.77	1.70	2.29	1.70	1.18
ν(CX);A_1_	1198	0.84	0.71	0.67	0.68	0.47	4.80	0.80	0.95
12, δ_ring_;A_1_	1004	1.00	1.00	1.00	1.00	1.00	1.00	1.00	1.00
1, ν_ring_;A_1_	784	0.26	0.19	0.11	0.27	0.11	2.72	0.45	0.46
6a, δ_ring_;A_1_	468	0.18	-	0.43	0.31	0.36	0.18	1.70	0.70

^1–3^: Intensity ratio measured from the Raman of the aqueous solution (Sol.) and the SERS spectra recorded without ^2^ and with ^3^ chloride ions, respectively.

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
