# Peer review of "A DFT Approach to the Surface-Enhanced Raman Scattering of 4-Cyanopyridine Adsorbed on Silver Nanoparticles"

_nanomaterials, 2019, doi:10.3390/nano9091211_

Round 1

Reviewer 1 Report

The manuscript “A DFT Approach to the Surface-Enhanced Raman Scattering of 4-Cyanopyridine Adsorbed on Silver Nanoparticles” reports about DFT studies of the SERS effect between silver nanoparticles and 4-cyanopyridine monomer. The data are clearly presented and the reported conclusions are justified on the basis of accurate theoretical studies.

My main concerns are about the experimental part of the manuscript that lacks of details and information. In particular:

The silver nanoparticles preparation procedure is usually crucial to obtain plasmon nanoparticles able to enhance some Raman signals. It depends on many parameters and in particular on the shape (sphere, ellipse, rods?) and size of the nanoparticles. The authors just reported the UV-Vis spectrum of plasmon NPs but the spectrum is not discussed to give information about size and shape. For example, the figure S1 clearly shows the appearance of a new signal at about 570 nm, that could be due to elongated nanostructures (or aggregates?). In this context, further characterizations such as DLS or morphological characterization should be useful to confer reproducibility to the study. What do the authors mean with “Raman spectra of … 0.7 M aqueous solution”? They acquired the Raman spectrum of a cast film from water dispersion/solution or they did it directly onto the suspension/solution? In first case, how many regions of the film were investigated? Are the spectra intensity reproducible? Is it possible to report the standard deviation of the measurements? Anyway, caption of Figure 1 has to be improved.

Reviewer 2 Report

The manuscript by Lopez-Tocon et al. reports characterization of 4-cyanopyridine bonding to silver surface based on DFT and SERS studies. This is a very nice manuscript clearly demonstrating power of coupled DFT and SERS approach for analysis of subtle details in organization of surface complex. The manuscript is very well-written and clearly presented. I recommend publication of this work in NANOMATERIALS after minor revision considering the following points:

1) In this work metal surface was modeled by one or two metal atoms/ions; this is very approximate approach. Usually in such kind of DFT studies more metal atoms were employed for construction of metal cluster, especially in the case if surface charge is important. Please discuss this matter in the revised version of manuscript.  

2) Please indicate in the revised version of manuscript peak positions of SERS bands which are indicators of the reduction of nitrile group.

3) In this work high concentration of chloride ions was used. Thus, part of silver nanoparticles surface is occupied by adsorbed Cl- anions. This affects both interaction between the adsorbed 4-cyanopyridine molecules and interfacial electrochemical potential. Please discuss this matter in the revised version of manuscript.

Round 2

Reviewer 1 Report

The authors properly amended the manuscript according to the proposed suggestions. The manuscript is now suitable for publication in a Nanomaterials.

Reviewer 2 Report

The questions raised by the referee are carefully addressed and the revised version of manuscript is acceptable for publication in NANOMATERIALS in present form.